# Immunohistochemical Expression of Vitamin D Receptor in Uterine Fibroids

**DOI:** 10.3390/nu14163371

**Published:** 2022-08-17

**Authors:** Anna Markowska, Paweł Kurzawa, Wiesława Bednarek, Anna Gryboś, Marcin Mardas, Monika Krzyżaniak, Jan Majewski, Janina Markowska, Marian Gryboś, Jakub Żurawski

**Affiliations:** 1Department of Perinatology and Women’s Diseases, Poznan University of Medical Sciences, 60-535 Poznan, Poland; 2Department of Clinical Pathology and Immunology, Poznan University of Medical Sciences, 60-355 Poznan, Poland; 3Department of Oncological Pathology, Hospital of Lord’s Transfiguration, Partner of Poznan University of Medical Sciences, 61-848 Poznan, Poland; 4I Department of Gynecological Oncology and Gynecology, Medical University of Lublin, 20-059 Lublin, Poland; 5Department of Gynecology and Obstetrics, Faculty of Health Sciences, Wroclaw Medical University, 50-367 Wroclaw, Poland; 6Department of Gynecologic Oncology, Division of Oncology, Poznan University of Medical Sciences, 60-535 Poznan, Poland; 7Institute of Health Sciences, University of Opole, 45-040 Opole, Poland; 8Department of Immunobiology, Poznan University of Medical Sciences, 60-806 Poznan, Poland

**Keywords:** uterine fibroids, vitamin D, vitamin D receptor (VDR)

## Abstract

One of the many factors involved in the development of uterine fibroids is vitamin D deficiency. One aspect of this deficiency is decreased serum concentration of calcidiol-25(OH)D, a metabolite of D3 vitamin. The active form of vitamin D3, which arises after numerous enzymatic reactions, is calcitriol-1,25(OH)2D3; this compound is transported to various body tissues. Vitamin D possesses extra-genomic effects due to its influence on various signaling pathways, i.e., through activating tyrosine kinases and by genomic effects via binding to a specific nuclear receptor, vitamin D receptor (VDR). The vitamin D/VDR complex regulates the expression of genes and is involved in the pathogenesis of fibroids. Numerous studies have shown that vitamin D supplementation reduces fibroid size. It has also been shown that the expression of VDR in myoma tissue is significantly lower than in the uterine muscle tissue at the tumor periphery. However, the expression of VDR in non-myoma uterine muscle has not previously been investigated. Our VDR expression studies were performed immunohistochemically with tissue microarrays (TMA) in three tissue groups: 98 uterine myoma tissues, 98 uterine tissues (tumor margin), and 12 tissues of normal uterine muscle (i.e., without fibroids). A statistical analysis showed significantly lower VDR expression in uterine muscle at the periphery of the fibroid than in healthy uterine muscle. Lower expression of VDR at the periphery of the myoma compared to that in normal uterine muscle may indicate potential for new myomas. This observation and the described reduction in the size of fibroids after vitamin D supplementation supports the hypothesis of causal development of uterine fibroids and may be useful for the prevention of re-development in the event of their excision from the uterus.

## 1. Introduction

Uterine fibroids are benign monoclonal tumors that occur in more than 70% of women of childbearing age. Notably, 25–50% of cases are asymptomatic. However, symptoms of uterine fibroids include heavy menstrual bleeding resulting in iron deficiency anemia, pressure symptoms on adjacent organs (bladder and intestines), decreased fertility, miscarriages and complications during childbirth [1,2,3].

Many factors are involved in the etiology and development of uterine fibroids, including steroid hormones, i.e., estradiol and progesterone, extracellular matrix (ECM), microRNA, genetic factors, stem cells, cytokines and growth factors [4,5,6,7,8,9,10]. Research indicates that vitamin D deficiency may be an important risk factor in fibroid development [11,12].

The participation of estrogens in fibroid development is limited to fibroids produced by the ovaries or by the conversion of androgens from the adrenal glands and ovaries, influenced by CYP19 aromatase, an enzyme from the cytochrome P450 family. Through nuclear α and β receptors, estrogens stimulate the secretion of numerous growth factors and sensitize myoma tissue to the action of progesterone, which induces myoma development by regulating genes controlling proliferation and apoptosis [4,5]. Both estradiol and progesterone affect the production and accumulation of ECM, mainly by increasing the expression of fibronectin and collagen, thereby causing tumor fibrosis, which is a component of the myoma background [5,6,13].

Studies by Cordozo et al. [7] have shown that miR-21 (small, non-coding RNA) participates in the development of fibroids by influencing ECM mediators such as TGF-β3 (transforming growth factor β3) and MMPs (metalloproteinases), taking part in the formation of ECM.

Genetic analyses revealed that the development of fibroids was influenced by mutations in the MED12 gene, located on chromosome 13. Regulated by signaling the canonical WNT/β-catenin pathway, these mutations are present in approximately 50–75% of myomas. In over 60% of cases, these mutations are accompanied by overexpression of high mobility group protein (HMGA) with high electrophoretic mobility, which is statistically significantly more common in fibroid tissue than in the surrounding myometrium [8,14,15,16].

Stem cells have been shown to be involved in the initiation and development of fibroids. This small subpopulation of cells with unique characteristics, including self-renewal and asymmetric division, is found in the endometrial and myometrial layers. Such cells act mainly through the WNT/β-catenin pathway, as well as through IGF-2 (insulin-like growth factor 2) and IR-A (insulin receptor A) [9,15,17]. According to Moravek et al. [18], these cells are essential for steroid-dependent fibroid growth.

Research results indicate the role of vitamin D deficiency in developing uterine fibroids [11,12,13,19,20,21]. Vitamin D is a fat-soluble organic steroid compound occurring in several forms (vitamin D1-calciferol, vitamin D2-ergocalciferol and vitamin D3-cholecalciferol). Its sources are either synthesis in the skin under the influence of sunlight (UVB) with a wavelength of 290–315 nm or food containing this vitamin. After absorption into the blood, it binds to the transport D-binding protein (DBP). As a result of enzymatic reactions (including CYP 2R1-cytochrome P450 hydrolase) in the body’s tissues (mainly in the liver and kidneys), an active metabolite is formed, i.e., calcitriol-1,25 dihydroxy-cholecalciferol: 1,25 (OH)2D3, which is transported to different tissues. Vitamin D has extra-genomic effects, by acting on tyrosine kinase activation in various signaling pathways, and genomic effects, by binding to a specific nuclear receptor, vitamin D receptor (VDR), which is present in various tissues. This complex influences the epigenome and the regulation of the expression of over a thousand genes present in various tissues of the organism [22,23,24,25].

In evaluations of the concentrations of the liver metabolites, hydroxyvitamin D-25 (OH) D-calcidiol, which includes 25-hydroxyvitamin D2 and hydroxyvitamin D3, is the best indicator of the state of the body’s supply of vitamin D due to its stability and long half-life. It also provides information about the biological availability of active calcitriol. According to the Endocrine Society of Clinical Practice Guidelines, a sufficient vitamin D concentration in the blood serum is ≥30 mg/mL [26]. According to more recent data, the preferred concentration is 40–60 mg/mL [24].

The pleiotropic mechanisms behind the ability of vitamin D to reduce the risk of uterine fibroid development and inhibit their growth are associated with its antiproliferative effects, induction of apoptosis, inhibition of angiogenesis and matrix metalloproteinases activity suppression. Anti-estrogen/anti-progesterone activity and the probable beneficial effect of the damaged DNA protein repair system via the D3/VDR axis have also been described [11,12,19,21].

The inhibitory effect of calcitriol [1,25 (OH)2D3] on the development of myoma in vitro was documented by Blauer et al. [27]. Different doses of calcitriol were added to myoma cells, and unchanged myometrium was obtained during surgery; high concentrations inhibited the proliferation of both cell types by 62%.

Clinical observations have confirmed the inhibitory effect of vitamin D on the development of uterine fibroids. A prospective, double-blind clinical trial in 69 women with uterine fibroids and diagnosed vitamin D deficiency showed a statistically significant beneficial effect of vitamin D supplementation (*p* < 0.001) [28].

A clinical trial with vitamin D supplementation in Chinese women to prevent and inhibit fibroid growth is currently underway [29].

Therefore, vitamin D supplementation is probably a protective option. According to Ciebiera et al. [12], the additional beneficial effects of using this vitamin may be significant.

## 2. Materials and Methods

The expression of VDR was tested on tissue in paraffin blocks from uterine specimens from the two groups of patients using the immunohistochemical method. All slides were stained with anti-Vitamin D3R antibodies.

Ninety-eight women with fibroids were included in the main group, i.e., the study group; they ranged from 24 to 82 years of age, with an average of 50 years. Twenty-five women had only one fibroid, another 6 had two fibroids and the rest (67) had three or more. The fibroids ranged from 1 cm to 10 cm in diameter, with an average of 4.3 cm in the biggest diameter. The study was performed on tissue arranged in microarrays from 98 uterine myoma and 98 uterine tissues (tumor margin). Each leiomyoma consisted of elongated smooth muscle cells without atypia, with very low mitotic rates, i.e., less than 1 mitosis/10 HPF. There was no necrosis or other regressive changes.

The control group comprised 12 women without fibroids, i.e., women which were operated on because of endometrial hyperplasia or genital prolapse. Samples from this group comprised normal uterine muscle. Subjects in this group ranged from 56 to 69 years of age, with an average of 61 years. The tissue fragments of normal myometrium included the tissue of tumor margins and that from the control group of patients. All samples were arranged in the three layers which are characteristic of a normal uterine wall.

For immunohistochemical VDR expression in the study group, 12 tissue microarray paraffin blocks (tissue micro array (TMA)) were performed. They contained 212 cores from 106 patients and 12 from the control material (spleen).

Areas of viable fibroid and tumor margin tissue elements were revived and marked by a pathologist for the microarrays. TMAs were assembled using a UNITMA Quick-Ray^®^ Manual Tissue Microarrayer.

For each 11 TMA paraffin blocks, two core samples of fibroid and tumor margin tissue were acquired from donor blocks.

Initial sections were stained for hematoxylin and eosin to verify the histopathological findings. While preparing the slides, eight of the cores fell out; as such, the final number of cases in the study group was 98 (Figure 1 and Figure 2).

### 2.1. Immunohistochemical Studies of Tissue Microarrays

Serial four-micrometer tissue sections were cut from the TMA blocks containing cores of fibroid and tumor margins and applied to special coated slides. Rabbit polyclonal antibody to Vitamin D3 Receptor (Biorbyt orb214726) was used to demonstrate the antigens present in the tissue material (Figure 3). Sections treated with a matched concentration of non-immune IgG were used as a negative control. Spleens were used as a positive control for immunohistochemical staining and localization.

Slides were incubated in a water bath at 96 °C in citrate buffer at pH 6.0 for 50 min. Endogenous peroxidase activity was blocked with 3% H_2_O_2_.

Tissue material was incubated with the ImmPRESS^®^ Horse Anti-Rabbit IgG PLUS Polymer Kit -ImmPRESS Horse Anti-Rabbit IgG Polymer Reagent and BLOXALL^®^ Blocking Solution (Vector Laboratories Inc., Burlingame, 94010 CA, USA, Catalog no.: MP-7801) for 30 min. The antigen was localized using Chromogen DAB-3.3 in all preparations. The antibody stained the cell nuclei in brown. The membrane and cytoplasm of each cell were unstained (Figure 3).

### 2.2. Light Microscopy Techniques for Cell Imaging

Photographs of the tissue microarrays were taken, including sections of fibroids, tumor periphery and a control group, using an Olympus BX 43 light microscope with an XC 30 digital camera (Olympus, Tokyo, Japan). Magnification was set at 400×. Based on the obtained images from the light microscope, a semi-quantitative analysis of immunopositive cells was performed. Calculations were made using the Olympus cellSens commercial software. Phase analysis of immunohistochemically stained tissue microarrays was undertaken, including automatic detection of objects based on their color, shade intensity or shape. In this case, the color criterion—the browning of the cell nuclei as a result of DAB staining—was selected. The computer program automatically classified cells based on predefined threshold values. The data were exported to MS Excel files and used for further statistical analyses [30].

### 2.3. Statistical Analysis

Statistical analyses were performed in the R program, version 3.5.1 (R Core Team (2018). R: Language and environment for statistical computing by R Foundation for Statistical Computing, Vienna, Austria). The normality of the distribution of the number of immunopositive cells was checked using the Shapiro-Wilk test and indicators of skewness, kurtosis and visual assessment of histograms. A comparison of the mean number of immunopositive cells between the tumor periphery and the myoma itself was performed using the student’s t-test for paired measurements and the control group. The mean difference (MD) with a 95% confidence interval (CI) was calculated. Bonferroni correction was applied due to multiple comparisons. A significance level of α = 0.05 was used.

The study was approved by the Bioethics Committee of the Medical University of Karol Marcinkowski in Poznań on 16 January 2022.

## 3. Results

The mean number of immunopositive cells was significantly greater in the peripheries of tumors than in the myomas themselves (MD = 190 (95%: 80; 300), *p* = 0.004). A significant difference was also confirmed in the number of immunopositive cells in the peripheries of the myomas and in the myomas themselves compared to the control group (MD = −434 (95%: −672; −200), *p* = 0.004 for tumor periphery and MD = −624 (95%: −861; −386), *p* < 0.001 for fibroid) (Figure 4, Table 1).

## 4. Discussion

Vitamin D is a fat-soluble compound that can occur in several forms (vitamin D1-calciferol, vitamin D2-ergocalciferol and vitamin D3-cholecalciferol). There are two sources of Vitamin D for humans: food or synthesis in the skin due to sun exposure. Along with vitamins A, C and E, it is an antioxidant agent [31].

Vitamin D can affect cells in two ways: non-genomic and genomic one. The latter occurs through the nuclear vitamin D receptor (VDR), which is associated with the regulation of the expression of multiple genes. Numerous studies have demonstrated the anticancer activity of vitamin D due to pleiotropic action involving the following mechanisms: inhibition of cell proliferation, inhibition of angiogenesis, activation of apoptosis and suppression of metalloproteinase activity, which is associated with the inhibition of cell mobility by adhesion proteins [32,33].

Using immunohistochemical methods, this study was designed to assess the VDR which leads to the pathogenesis of fibroids. Tissue was collected from two group of patients. The study group included 98 women. Two tissue fragments were collected from each patient: one comprising fibroid tumor and the second from the tumor periphery. The control group consisted of 12 patients whose tissue came from unchanged myometrium.

Microscopically, all tissue fragments consisted of smooth muscle cells which did not reveal any atypia and did not show any significant morphological changes among normal myometrium, tumor periphery tissue and tumor cells. The only microscopic change was found in the distorted architecture of tumor tissue.

Immunohistochemically, this study discovered the highest number of immunopositive cells in normal tissue, a lower number in the tumor periphery and the lowest number in the tumor cells. Lima et al. conducted a cross-sectional controlled study of 40 women who had undergone abdominal hysterectomy to compare the immunohistochemical expression of VDR in uterine leiomyoma tissue samples with that in adjacent nonneoplastic myometrial tissue. The study results showed a reduced expression of VDR in the uterine leiomyoma compared to nonneoplastic myometrial tissue. The reason for the reduced expression of these receptors is still unknown, and it is unclear whether it is related to the initiation or progression of uterine leiomyomas [34]. In our group of 98 patients, we also found that the mean number of immunopositive cells was significantly larger in the peripheries of tumors than in the myomas themselves. Similarly, Halder et al. found reduced of VDR levels in over 60% of uterine tumors compared to myometrium. Additionally, however, a semi-quantitative analysis of the immunohistochemical reaction showed a higher mean number of VDR immunopositive cells in the tumor-unchanged tissue [35]. Increased proliferation and deregulation of the Wnt/β-catenin pathway are important in the formation and growth of leiomyomas. Vitamin D has an antiproliferative effect on HULP cells by arresting cell growth and inhibiting the Wnt/β-catenin pathway. Lower VDR expression may reducing the antiproliferative activity of vitamin D, thereby contributing to the formation of uterine leiomyomas [36].

Vitamin D reduces the growth of fibroids, not only by likely inhibiting the activation of the Wnt/β-catenin and TGF*β*3 pathways, but also through its involvement in the DNA repair network [37].

Based on the above data and the results of our research, we suggest that there is a relationship between decreased VDR expression and the development of human uterine fibroids.

The results of the numerous studies presented above confirm that vitamin D deficiency is associated with fibroid pathogenesis and that supplementation has a positive effect on the reduction of their size [11,12,19]. However, several studies involved small groups of patients. A case-control study (30 women in total) also showed that the effect of vitamin D supplementation was enhanced with vitamin B6 and plant-derived polyphenol (epigallocatechin gallate), which may influence assessments of vitamin D activity alone [38].

In their review, Vergara et al. [39] suggested the importance of ethnicity on serum concentrations of vitamin D (significantly higher levels of vitamin D deficiency were found in African-American women compared to Caucasian women) and genetic polymorphisms related to vitamin D metabolism.

Therefore, it seems that the determination of vitamin D–VDR expression, a member of the nuclear receptor family found in nearly all tissue, may be a more accurate means of detecting local vitamin D deficiency than its serum concentration. Furthermore, the vitamin D/VDR complex regulates the expression of genes also involved in myoma pathogenesis [22,23,39].

A previous VDR study showed that only 25 women (60%) had a reduction in VDR expression in myoma cells compared to the adjacent healthy tissue. In that research, VDR was not studied in women without uterine fibroids. The studied patients belonged to various ethnic groups [35]. The results of other studies also indicated VDR expression in unchanged tissue in women with fibroids [21,35,39,40].

In our study, patients were ethnically homogeneous. We determined the expression of VDR in the uterine muscle of women without fibroids (control group). It was revealed that the expression of VDR was significantly lower in the myometrium in the peripheries of fibroids than in the uterine muscle in the control group.

This significant relationship indicates the potential for new fibroid formation in women with diagnosed uterine fibroids. This may be related to the redevelopment of fibroids in women who have been operated on for this reason (e.g., after enucleation). This insight and the results of serum vitamin D levels support the application of conservative treatment of uterine fibroids.

## 5. Conclusions

Lower VDR expression in uterine muscle in the peripheries of fibroids compared to that in non-fibroid uterine muscle may be a potential cause of the development of new fibroids.

## Figures and Tables

**Figure 1 nutrients-14-03371-f001:**
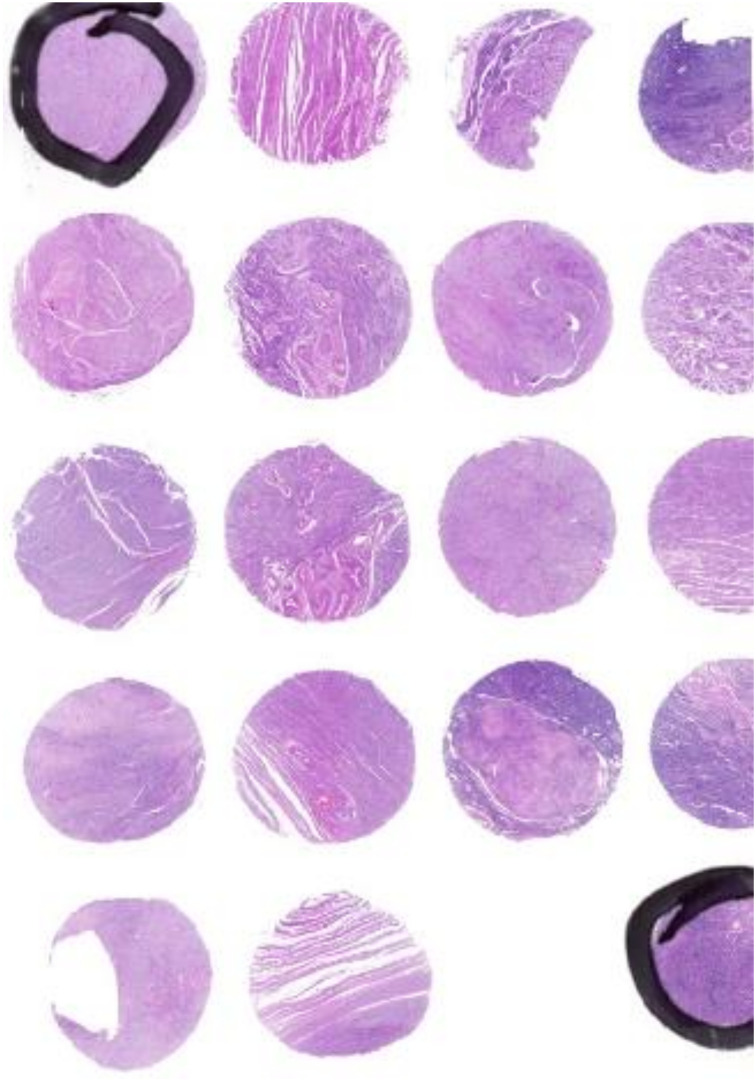
Tissue micro array of fibroid and tumor margin, stained with H&E, 100×.

**Figure 2 nutrients-14-03371-f002:**
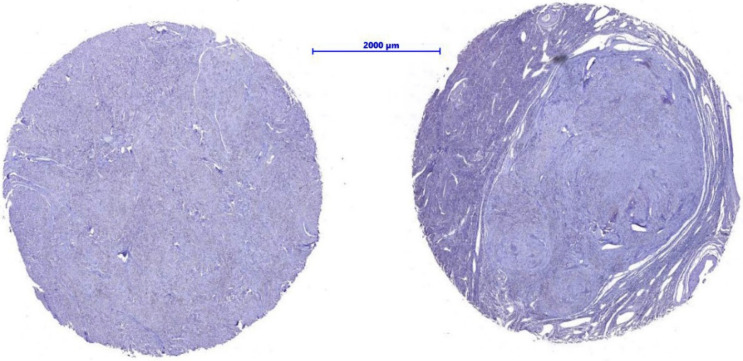
Example of fibroid and tumor margin, stained with H&E, 250×.

**Figure 3 nutrients-14-03371-f003:**
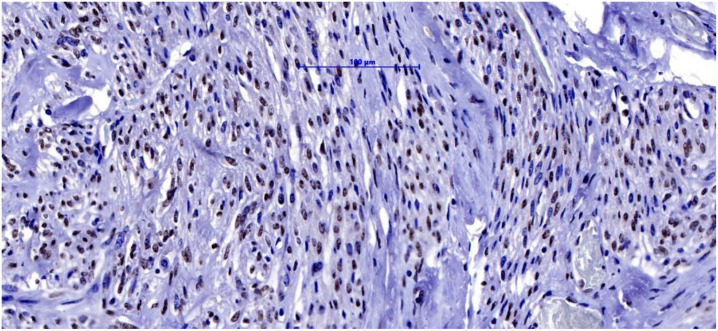
The nuclei of the myometrium stained with Vit. D3 Receptor antibody, 300×.

**Figure 4 nutrients-14-03371-f004:**
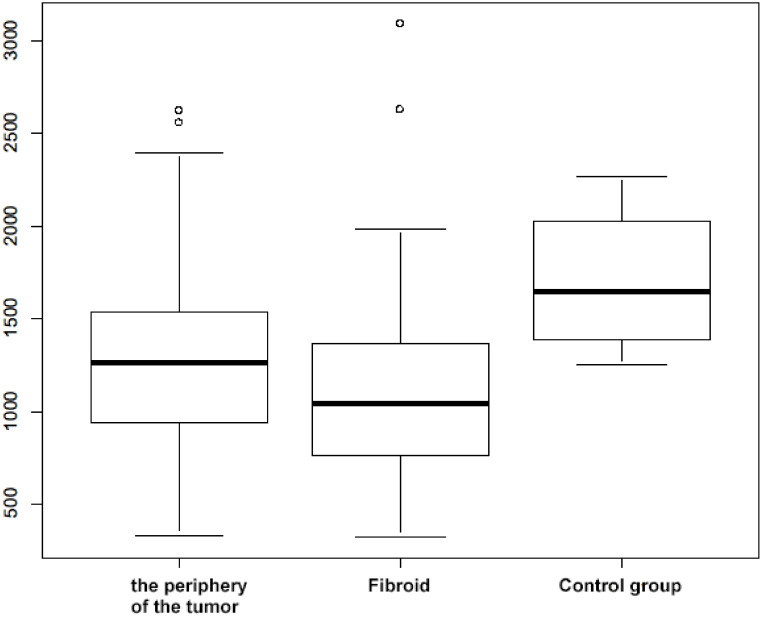
Boxplot of the number of immunopositive cells in both groups, Circles indicate outlier values (i.e. values outside 1.5 times the interquartile range above the upper quartile and below the lower quartile: Q1 − 1.5 × IQR or Q3 + 1.5 × IQR, where IQR—interquartile range).

**Table 1 nutrients-14-03371-t001:** The number of immunopositive cells—comparison between groups.

	Tumor Periphery	Fibroid	Control Group	Tumor Periphery vs. Fibroid	Tumor Periphery vs. Control Group	Fibroid vs. Control Group
*n*	98	98	12			
Mean ± SD	1300 ± 500	1100 ± 500	1700 ± 350	t = 3.32 df = 97 *p* = 0.004	t = −3.86 df = 16 *p* = 0.004	t = −5.56 df = 16 *p* < 0.001
Range	330 to 2622	321 to 3092	249 to 2271			

## Data Availability

Not applicable.

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
