# Peer review of "Immunohistochemical Expression of Vitamin D Receptor in Uterine Fibroids"

_nutrients, 2022, doi:10.3390/nu14163371_

Round 1

Reviewer 1 Report

This is a straightforward study to determine the VD receptor expression in uterine fibroid tissues. Although the results are interesting to some levels, the overall data are not convincing to show either VD is deficient in these samples or VD receptor is decreased. Such results need to be further reinstated by combination of multiple approaches, in order to achieve conclusion.

Author Response

1. Point 1: This is a straightforward study to determine the VD receptor expression in uterine fibroid tissues. Although the results are interesting to some levels, the overall data are not convincing to show either VD is deficient in these samples or VD receptor is decreased. Such results need to be further reinstated by combination of multiple approaches, in order to achieve conclusion.

 Response 1: In line with the Reviewer's observation, this is actually quite a simple study aimed at determining the immunohistochemical expression of the vitamin D receptor in uterine fibrous tissues. However, there are no such data in the literature regarding the semi-quantitative analysis of the results of immunohistochemical studies in uterine fibroids, both in the center and in the periphery in relation to the control group.

In addition, the material tested came from several gynecological centers and its development was very laborious. The results were obtained from the morphometric analysis of several hundred photomicrographs.

Thank you for reading our manuscript carefully. We are also grateful for all suggestions.

Yours faithfully,

Reviewer 2 Report

Markowska et al. reports the immunohistochemical expression of vitamin D receptor with tissue microarrays in 98 uterine myomas compared to 98 uterine myomas margins (periphery of the tumor) and 12 normal uterine muscle. The rationale behind this study is that vitamin D receptor complex regulates the expression of genes involved in the pathogenesis of fibroids with a different expression between myoma and adjacent normal tissue.

Although the principal result of the study which is the lower expression of vitamin D receptor in myoma tissue is not novel, the authors add data about the expression of vitamin D receptor in non-myoma uterine muscle tissue on the tumor periphery.

Major Comments:

1.     The introduction is too long, the authors should shorten it listing all the factors involved in the pathogenesis of myomas in a table

2.     Materials and Methods are too long, the authors should shorten it listing the different steps for setting up the tissue microarrays in a table or in a figure

Minor Comments:

1.     The authors should specify everywhere in the manuscript that the normal uterine tissue at the periphery of the tumor (representing the comparison group) is represented by normal uterine muscle tissue

Author Response

1. Point 1: The introduction is too long, the authors should shorten it listing all the factors involved in the pathogenesis of myomas in a table.

 Response 1: In agreement with the reviewer's suggestion, we finalized the "introduction". Unfortunately, it is difficult for us to respond to the Reviewer's suggestion regarding the presentation of all factors involved in the pathogenesis of myomas should be included in the table. We believe that each pathogenetic mechanism is described in detail and the various differences between them would not fit into the table format. We've made some minor linguistic fixes that should make reading easier.

2. Point 2: Materials and Methods are too long, the authors should shorten it listing the different steps for setting up the tissue microarrays in a table or in a figure.

Response 2: The section "Materials and methods" has been shortened as suggested by the Reviewer. We have presented only the most important stages of creating microarrays and performing immunohistochemical studies. 3.     

3. Point 3: The authors should specify everywhere in the manuscript that the normal uterine tissue at the periphery of the tumor (representing the comparison group) is represented by normal uterine muscle tissue. 

Response 3: We improved the text in certain places.

Thank you for reading our manuscript carefully. We are also grateful for all suggestions.

Yours faithfully,

Round 2

Reviewer 1 Report

I still feel the quantitive data solely presented by a single experiment is insufficient. Although samples were collected from different centers. The research results were more suitable to submit to data center rather than publishing a research article.

Author Response

Point 1: English language and style are fine/minor spell check required.

 Response 1: The spelling has been checked and errors have been corrected.

Thank you very much for revising our manuscript. We appreciate all suggestions. Thank you also for your kindness.

Yours faithfully,

Reviewer 2 Report

Congratulations to the authors, the work can be accepted in this form, I have no other comments to add 

Author Response

Point 1: Congratulations to the authors, the work can be accepted in this form, I have no other comments to add 

 Response 1: Dear Reviewer,

Thank you very much for revising our manuscript. We appreciate all suggestions. Thank you also for your kindness.

Yours faithfully,